



# Aerosol optical depth climatology from the high–resolution MAIAC product over Europe: differences between major European cities and their surrounding environments

Ludovico Di Antonio[1], Claudia Di Biagio[2], Gilles Foret[1], Paola Formenti[2], Guillaume Siour[1], Jean–François Doussin[1], and Matthias Beekmann[2]

[1]Univ Paris Est Creteil and Université Paris Cité, CNRS, LISA, F–94010 Créteil, France
[2]Université Paris Cité and Univ Paris Est Creteil, CNRS, LISA, F–75013 Paris, France

*Correspondence to*: Ludovico Di Antonio (ludovico.diantonio@lisa.ipsl.fr), Matthias Beekmann (matthias.beekmann@lisa.ipsl.fr)

**Abstract.** The aerosol optical depth (AOD) is a derived measurement useful to investigate the aerosol load and its distribution at different spatio–temporal scales. In this work we use long–term (2000–2021) MAIAC (Multi–Angle Implementation of Atmospheric Correction) retrievals with 1 km resolution to investigate the climatological AOD variability and trends at different scales in Europe: a continental (30–60°N; 20°W–40°E), a regional (100x100 km$^2$) and an urban local scale (3x3 km$^2$). The AOD climatology at the continental scale shows the highest values during summer (JJA) and the lowest during winter (DJF) seasons. Regional and urban local scales are investigated for twenty–one cities in Europe including capitals and large urban agglomerations. Analyses show AOD average (550 nm) values between 0.06 and 0.16 at the urban local scale, while also displaying a strong north–south gradient. This gradient corresponds to a similar one in the European background, with higher AOD being located over the Po–Valley, the Mediterranean basin, and Eastern Europe. Average enhancements of the local with respect to regional AOD of 57%, 55%, 39% and 32% are found for large metropolitan centers such as Barcelona, Lisbon, Paris and Athens respectively, suggesting a non–negligible enhancement to the aerosol burden through local emissions. Negative average deviations are observed for other cities, such as Amsterdam (–17%) and Brussels (–6%) indicating higher regional background signal and suggesting a heterogeneous aerosol spatial distribution that conceals the urban local signal. Finally, negative statistically significant AOD trends for the entire European continent are observed. A stronger decrease rate at the regional scale with respect to the local scale one occurs for most of the cities under investigation.

## 1. Introduction

Climate change and air quality preservation represent two of the greatest challenges of our times, especially in densely populated areas. Aerosol particles have been shown to play a key role in climate change and to affect air quality over many regions of the world (Robotto et al., 2022; Viana et al., 2014; Fiore et al., 2012). Aerosols affect the radiative budget both directly, by scattering and absorption of solar and thermal radiation (the aerosol radiation–interactions, ARI) or indirectly, by influencing the cloud formation and properties (aerosol–cloud interactions effect, ACI) (Bellouin et al., 2020). Constraining the aerosol contribution to climate and its change is still a challenge (Bender, 2020) as further demonstrated by the Climate Change 2021 IPCC report indicating still huge spread in ARI and ACI estimations (Masson-Delmotte et al., 2021). Atmospheric aerosols are also a concern for air quality and human health (Yang et al., 2018; Li et al., 2017, 2016; Dockery, 2009). Millions of people in Europe and around the world, especially over dense urban agglomerations, industrial areas and rural environment, are everyday exposed to significant aerosol levels (Sicard et al., 2021). Under favorable weather conditions such as high radiation levels, high temperature, low precipitations and low winds during summer, or temperature inversions



and low planetary boundary layer height during winter, primary and secondary aerosol local formation have been shown to build up to create the so called "aerosol pollution episodes" (Foret et al., 2022; Groot Zwaaftink et al., 2022; Diémoz et al., 2019). These episodes correspond to daily average PM levels above the European threshold of 50 μg m$^{-3}$ and lasting several consecutively days. If such episodes occur frequently, they lead to significant air quality and visibility degradation (Majewski et al., 2021; Singh et al., 2020) and increase the potential health risk (Luo et al., 2021; Grigorieva and Lukyanets, 2021).

However, the aerosol anthropogenic precursors, abundant in urban agglomerations, can also spread around emission hot spots and affect larger areas, including rural and forested environments, leading to situations of mixed anthropogenic–biogenic scenarios (Xu et al., 2021). This would lead aerosols to have different chemical, physical and radiative properties and therefore potential different impact both on human health (Tuet et al., 2017; Liu et al., 2009) , and the environment (Nascimento et al., 2021; Shrivastava et al., 2019; Martin et al., 2016). In this regard, how the local and regional scale anthropogenic and biogenic

precursors, their mixing and their processing affect aerosol loading and properties, in particular around major city agglomerations, is still unknown and matter of scientific investigation (Cantrell and Michoud, 2022; Liu et al., 2021; Ma et al., 2021).

The Aerosol Optical Depth (AOD) is a key parameter to investigate aerosol load, properties and distribution over local to large scale areas (Bai et al., 2022; Faisal et al., 2022; Raptis et al., 2020; Sun et al., 2019; Just et al., 2015; Smirnov et al., 2002).

The AOD is defined as the integral of the aerosol extinction coefficient (units of length$^{-1}$) over the whole atmospheric column and it depends on the aerosol mass concentration, size distribution, shape and complex refractive index. Measurements of AOD are used to improve the air quality forecasts since they can be assimilated in regional or global models (Lee et al., 2022; Ha et al., 2020; Kondragunta et al., 2008) and they can be also linked to visibility measurements (Aman et al., 2022; Zhang et al., 2016; Boers et al., 2015; Kessner et al., 2013; Bäumer et al., 2008). Moreover, the AOD  spectral variability can also be

used to discern among different aerosol types and help source apportionment (Tuccella et al., 2020; Bahadur et al., 2012). However, since AOD observations are vertically integrated, the correlation with surface aerosol measurements may not be straightforward (He et al., 2021; Grgurić et al., 2014; Segura et al., 2017; Guo et al., 2009; Schaap et al., 2009; Schäfer et al., 2008). In fact, AOD is sensitive to dust and biomass burning plumes transported at high altitude, which may not affect surface measurements (Eck et al., 2023; Gkikas et al., 2022; Song et al., 2009). Different studies reported AOD trends on a global

scale (Gupta et al., 2023, 2022; Zhao et al., 2017; He et al., 2016; Mehta et al., 2016; Mao et al., 2014) supporting a decreasing AOD trend over Europe (Gupta et al., 2023, 2022; Filonchyk et al., 2020b; Alpert et al., 2012). The overall decreasing trend at the European regional scale has been attributed to mitigation policies applied in recent years for the aerosol and the aerosol precursor emissions (Gupta et al., 2022; Provençal et al., 2017; Zhao et al., 2017).

The AOD is routinely retrieved across the globe by both ground−based sun photometers measurements, such those of the

widespread AERONET network (Aerosol Robotic Network) (Giles et al., 2019), and by satellite sensors, among them the Moderate Resolution Imaging Spectroradiometer (MODIS). Three complementary algorithms, developed at NASA, exist for the MODIS aerosol AOD retrieval:  the Deep Blue (DB) (Hsu et al., 2004), the Dark Target (DT)(Remer et al., 2020, 2005) and the more recent Multi–Angle Implementation of Atmospheric Correction (MAIAC) algorithm (Lyapustin et al., 2018). The DB and DT algorithms, extensively used in literature (e.g., Shi et al., 2021; Spencer et al., 2019; Sayer et al., 2018; Lee

et al., 2017; Hsu et al., 2017), provide aerosol retrievals at the spatial resolution of 3km and 10 km. The MAIAC algorithm provides atmospheric retrievals of AOD at 470 and 550 nm at the more advanced spatial resolution of 1 km. As a matter of fact, an accurate estimation of surface reflectance, discerning among atmospheric and surface contributions, is necessary to provide the best quality AOD retrievals (Bilal et al., 2019). In this regards, the MAIAC algorithm benefits of the multi–angle satellite observations, retaining in memory up to 16 days of consecutive satellite overpasses, to better constrain the surface

reflectance, improving the AOD retrievals in particular over complex scenes as urban areas (Chen et al., 2021; Gupta et al.,



2016; Wang et al., 2010). The MAIAC aerosol algorithm uses eight different background aerosol models over land (Look Up Tables, LUT) and it has developed a more stable algorithm that reduces the AOD bias over bright surfaces (in absence of smoke and dust), typical for the DT and DB algorithms (Lyapustin et al., 2018). Furthermore, MAIAC can retrieve AOD over partial cloudy conditions and distinguish between smoke and dust scenes. The AOD from the MAIAC algorithm has been

validated over different areas of the world and shown to perform better with respect to the DT and DB algorithms when compared to AERONET observations (Falah et al., 2021; Qin et al., 2021; Martins et al., 2019; Zhang et al., 2019; Tao et al., 2019; Mhawish et al., 2019; Martins et al., 2017; Just et al., 2015). The estimated expected error (EE) for MAIAC AOD is evaluated at $\pm(0.05 + 0.1AOD)$, but it is shown to vary as a function of surface reflectivity, aerosol loading and size, as well as aerosol type (Falah et al., 2021). Because of its 1 km resolution and good performances, the MAIAC AOD product has

increasing use in air quality studies (Pedde et al., 2022; Gladson et al., 2022; Yang et al., 2022; Jung et al., 2021; Hough et al., 2021).

In this paper, we benefit from the high–spatial resolution MAIAC long–term data (from 2000 to 2021) to investigate AOD over Europe. This work is part of the ACROSS (Atmospheric ChemistRy Of the Suburban foreSt, https://across.cnrs.fr/) project, whose objective is to deepen the current physical–chemical knowledge of the interaction between anthropogenic

emissions on the Paris area and its surrounding environment, through an intensive field campaign which took place in the summer 2022 (Cantrell and Michoud, 2022). Within the ACROSS context, this study wants to achieve three different objectives:

- Investigate the urban local vs regional scale aerosol optical depth variability starting from a broader context over the European domain (20°W–40°E,30–60°N) up to the urban local scale (3x3 km²) around major urban agglomerations

in Europe;
- explore the long–term trends at the urban local (3x3 km²), regional (100x100 km²) and continental scales (20°W–40°E,30–60°N);
- contextualize the results for the Paris agglomeration with respect to other European cities.

The manuscript is organized as follows. The MAIAC product and its use are described in Section 2. Previous validation studies

of the MAIAC product in Europe have been performed in Italy (Stafoggia et al., 2017), the Moscow metropolitan area (Zhdanova et al., 2020) and Germany (Falah et al., 2021), but no analysis have considered the entire European continent. Therefore, a validation analysis for Europe is also provided in Section 2. The discussion of the AOD climatology and trends over Europe and local/regional analysis will be presented and discussed in Section 3, before giving Conclusions in section 4.

## 2. Methods

### 2.1 MAIAC dataset extraction and analysis


The daily MCD19A2 product (Lyapustin and Wang, 2018) providing the AOD at 470 nm and 550 nm has been used over the period February 2000–August 2021. All the observations are delivered in the HDF4 format and stored at 1 km resolution in sinusoidal grid mapping. The product, distributed on a daily basis, contains the collection of each MODIS Aqua and Terra satellites overpasses, whose number varies according to the latitude. In this product, in order to merge the satellite data to

perform the climatological averages at the European scale (20°W–40°E,30–60°N), the daily average of each tile has been taken, followed by horizontal and vertical concatenation over the different MODIS tiles of interest. Only data classified as best–quality AOD (quality check flag "0000") have been used in the following analysis. Although this choice reduces the



number of available data, it guarantees the quality of the retrieval which is an important aspect to perform high resolution studies over urban areas.

Starting from the merged MAIAC data, the following treatment is applied:

- Sinusoidal to WGS 84 grid coordinate system conversion.

- AOD daily averages are calculated for each grid point taking into account available observations in the day from Terra and Aqua (i.e. 2 to 5 observations per day are available for the different grid points with Terra and Aqua overpasses times between 9 AM and 2 PM local time).

- Local and regional scale AOD extractions have been performed to investigate the effect of the aerosol formation and city emissions over the surrounding areas. To this aim a list of 21 cities has been established, including European capital cities and big agglomerates with more that 1 million inhabitants. Those cities are listed in Table 1 and their location is plotted in Figure 1. The MAIAC AOD data have been extracted around the city locations using two different concentric kernels (centered on the nearest pixel to the longitude and latitude values of each city in Table 1): 3x3 $km^2$ (9 $km^2$ area) for the local scale and 130 100x100 $km^2$ (10000 $km^2$) for the regional scale. The regional domain was chosen large enough in order to avoid effects due to the city and its plume, i.e. the local scale product occupies only ~0.09% of its regional background. Days for which a minimum of 40% spatial data coverage is available are considered, the others are discarded for the analysis. The local–to–regional AOD ratio (LTRR) has been calculated for each available kernel extraction to quantify the local scale enhancement to the regional AOD by using the following formula:

$$LTRR = \frac{AOD_{local}}{AOD_{regional}} - 1 \qquad (1)$$

Positive deviations of the LTRR highlights the positive contribution of the urban local scale to the regional background signal, considering that $AOD_{local}$ intrinsically represents the sum of the local production and the possible regional advected AOD fractions. Conversely, negative deviations can be linked to the presence of a non–homogenous spatial aerosol distribution at the regional scale, as well as to a possible local sink of pollution. The former may result in a stronger regional background 140 signal related to different aerosol sources surrounding the city which may conceal the urban local signal and reduce the pollution gradients.

Trend assessment on AOD has been conducted over annual averages of daily AOD data using the Original Mann–Kendall test (Hussain and Mahmud, 2019). Annual AOD averages are performed if at least 50 AOD daily data are available in the year, and trend evaluations are performed if at least 5 years data are available in the dataset. The output of the Mann–Kendall test 145 provides the significance of the test (p–value) and the Theil–Sen slope (Theil, 1992; Sen, 1968). All the tests have been calculated assuming a significance level (α) of 0.05 and the trend is considered significant if p–value<α. The relative change has been calculated following (Colette et al., 2016):

$$RC(\% \; year^{-1}) = \frac{s}{y0} \qquad (2)$$

where s is the Theil–Sen slope and y0 is the first available year for the trend evaluation. The uncertainty attributed to the 150 MAIAC AOD retrievals has been defined through the expected error (EE) considering both absolute and relative errors by attributing an absolute error of 0.05 and a relative error of 0.1 following (Falah et al., 2021; Lyapustin et al., 2018). As discussed in the next Section, the validation against AERONET will be considered to reevaluate the EE over Europe and subsequently update the MAIAC uncertainty.



**2.2 Validation against AERONET observations and revised MAIAC estimated error (EE) for Europe**

The MAIAC AOD validation has been performed by comparing the 550 nm AOD with all the available acquisitions (207 sites) in the AERONET Version 3 ground–based sun photometers network over continental Europe (Giles et al., 2019). Version 3 Level 2 AERONET data have been used (last access: 16 May 2023). AERONET provides AOD measurements at four different wavelengths: 440nm, 675nm, 860nm, 1020nm. The AOD at 550 nm has been extrapolated by assuming a power law relationship with the Angstrom exponent $\alpha$ (Ångström, 1929; Schuster et al., 2006) calculated between 440 nm and 675 nm:


$$AOD_{550} = AOD_{675nm} \left(\frac{440}{675}\right)^{-\alpha} \tag{3}$$

$$\alpha = -\frac{\log\left(\frac{AOD_{440nm}}{AOD_{675nm}}\right)}{\log\left(\frac{440}{675}\right)} \tag{4}$$

Since the AERONET measurements are taken at different elevation angles depending on the sun elevation over the horizon, the measurements may be considered representative of a larger area around the point of acquisition (Chen et al., 2020; Schutgens, 2020; Kinne et al., 2013). In order to improve the meaningfulness against the AERONET observations, the MAIAC

AOD have been additionally extracted by taking an average area of 0.06°x 0.06° over the AERONET site, corresponding to ~7x7 km². Indeed (Falah et al., 2021) show that MAIAC – AERONET comparisons given similar results for boxes between 1x1 km2 and 9x9 km2. Furthermore, AERONET AOD data between ±1H of the satellite passage have been considered to compare with MAIAC. The uncertainty on the AERONET AOD is $\Delta AOD_{AERONET}$=0.02 linked to calibration uncertainty (Sinyuk et al., 2020). As the differences between MAIAC and AERONET observations are attributed entirely to MAIAC

uncertainty, the derived MAIAC expected error is conservative. Different statistical indicators have been calculated to evaluate the comparison between MAIAC and AERONET AOD data. Those the Mean Bias Error (MBE), the Normalized Mean Bias (NMB), the Root Mean Square Error (RMSE), and the fraction of data within a factor of two (FAC2), as defined below (N is the number of data points):

$$MBE = \frac{1}{N}(AOD_{MAIAC} - AOD_{AERONET}) \tag{5}$$


$$NMB = \frac{\sum(AOD_{MAIAC} - AOD_{AERONET})}{\sum AOD_{MAIAC}} \tag{6}$$

$$RMSE = \sqrt{\frac{\sum(AOD_{MAIAC} - AOD_{AERONET})^2}{N}} \tag{7}$$

$$FAC2\ (\%) = fraction\ of\ data\ satisfying\ 0.5 \leq \frac{AOD_{MAIAC}}{AOD_{AERONET}} \leq 2 \tag{8}$$

The correlation between $AOD_{MAIAC}$ and $AOD_{AERONET}$ has been evaluated through the Pearson correlation coefficient R. The slope and the intercept of the regression line have been calculated taking into account the uncertainty in both coordinates using

the York regression (York et al., 2004). The comparison between MAIAC and AERONET AOD at 550 nm for all available European AERONET measurements from 2000 to 2021 is shown in Figure 2. The overall validation performed considering the entire dataset (panel a) shows a slight underestimation of the AOD from MAIAC with respect to AERONET, with a MBE (–0.02) and a RMSE (0.06) values similar to those retrieved in previous validation studies (Chen et al., 2020; Lyapustin et al., 2018; Martins et al., 2017). The probability density function (PDF) of the MAIAC−AERONET absolute differences (panel b

in Figure 2), shows a mean value and a sigma of –0.02 and 0.06, respectively. 77% of the AOD retrievals fall in an



EE=±$(0.05AOD + 0.05)$, with a relative error lower than the validation EE=±$(0.1AOD + 0.05))$ from (Falah et al., 2021) accounting for observations in Northern Africa, California and Germany, but comparable to the EE envelope (~74% of points falling within the EE) obtained in (Qin et al., 2021) over the Köppen climate classification of normally humid and warm climate (Cf) region, including part of the European domain.

Since dependency of EE on aerosol type and size has been evidenced by (Falah et al., 2021) a further detailed validation depending on the Angstrom Exponent (AE) between 440 nm and 870 nm has been performed and presented in Figure 2 (panels c, d, e). The AE, combined with AOD, is an indicator of the particle type and size. AE values lower than 1 can be associated to coarse–mode aerosols (sea–salt and dust), whereas AE values higher than 2 to fine–mode aerosols (urban pollution and smoke) (Schuster et al., 2006). In this regards, three different classes depending on AE have been identified, respectively

referred to coarse, mixed and fine aerosol particles: AE<0.5, 0.5≤AE<1.5, AE≥1.5. The MAIAC validation shows an R value of 0.84 for the overall validation (panel a) comparable with the 0.85, 0.81 and 0.87 for the coarse, mixed and fine classes respectively. The validation for the mixed− and fine−dominated classes show a satisfactory accuracy of the product, with an MBE of –0.02 for both and 79% and 81% of the points, respectively, respectively within the envelope of EE=±$(0.1AOD + 0.05))$ from (Falah et al., 2021). However, for the coarse−mode the MBE (–0.08) and EE (46%) are

significantly lower with respect to the other two classes. In case of AOD<0.25 (84% of points in the coarse−mode validation), attributable to a marine−dominated aerosols scene, the EE is 51%, whereas for AOD≥0.25 (16% of points in the coarse−mode validation), attributable to dust−dominated aerosols, the EE is significantly lower than 1 sigma. As a matter of fact, a higher EE is needed to contain the 68% of the MAIAC–AERONET differences for the coarse−mode validation. (Qin et al., 2021) show as MAIAC regional background models seem to be affected by local aerosol properties as they are retrieved by

AERONET climatology, suggesting that, further improvements are needed in case of coarse−mode dominated classification.

In summary, results of the validation against AERONET suggest that the EE for MAIAC for observations over Europe between 2000 and 2021 can be estimated at EE=±$(0.05AOD + 0.05)$, lower than the EE estimated by (Falah et al., 2021). The total MAIAC AOD uncertainty has been therefore revised to take into account this new estimation.

## 3. Results and discussions

### 3.1 European scale analysis

Before looking at the fine scale variability of the major European cities (section 3.2), we address here the question of their AOD European background levels and their seasonal variation, as seen by multi–year MAIAC observations. We place our findings in the context of previous analysis mainly based on spatially less refined MODIS observations (Gupta et al., 2023; Filonchyk et al., 2020a; Wei et al., 2019), introducing an analysis based on two decades of data, extending and validating

studies performed on a shorter time periods.

The aerosol optical depth variability at European scale is shown in Figure 3, reporting seasonal averages, and in Figure S1 in the Supplementary Information reporting the monthly averages of the AOD at 550 nm. The summer (JJA) season shows the highest AOD values ranging between 0.12−0.22 in the 30°N−60°N band, whereas DJF shows the lowest AOD values ranging between 0.06−0.09. Figure S1 depicts maps of monthly AOD averages and shows maximum values between April and July,

and minimum between November and January. A North−South latitudinal gradient is present for all the seasons, as shown in Figure 3 and Figure S2, with maximum gradient during the summer (JJA) and minimum during the winter (DJF) season. According to Figure S2, seasonal AOD averages range between 0.06−0.11 and 0.09−0.22 and in the 55°N−60°N and 30°N−35°N bands, respectively.



These findings with the MAIAC dataset are broadly in line with a previous analysis of MODIS and MISR data (Gupta et al.,
2023, 2022; Filonchyk et al., 2020a; Mehta et al., 2016). A North–South AOD gradient over Europe has been also found in
other MODIS studies (Merdji et al., 2023; Floutsi et al., 2016; Israelevich et al., 2012; Barnaba and Gobbi, 2004). Averages
over Western Europe and for the 2007–2016 period, (Zhao et al., 2018) find a broad spring summer AOD maximum extending
from April to July around 0.2 for MODIS Aqua and Terra and around 0.15 for MISR, and a winter December and January
minimum between 0.08 and 0.10. (Ma and Yu, 2015) attribute simulated spring maximum over Southern France and Corsica
over the western Mediterranean basin, especially to sulfate and dust, while other primary aerosol species (sea salt, primary
carbonaceous aerosol) show lower contributions and a flat seasonal variation. For secondary aerosol as sulfates, these larger
AOD values during the late spring, summer season are attributed to stronger photochemical activity due to increased oxidant
capacity (enhanced OH and ozone levels), whereas the contrary is expected for the late autumn, early winter minima. However,
these authors did not take into account secondary organic aerosol, it can be expected to be maximum during summer caused
by the higher biogenic volatile organic compounds (BVOCs), and increased photochemical activity during this season (Gao et
al., 2022). Possible fire events can also affect summer AOD peaks over Europe since they are more frequent during this period
(European Commission et al., 2022; Zhao et al., 2018).

Different aerosol hotspots, as previously identified in the literature (Coelho et al., 2022; Backes et al., 2016; Gkikas et al.,
2016; Bovchaliuk et al., 2013; Vecchi et al., 2009) are also visible in Figure 3, especially the Mediterranean Sea, the Po Valley
and Eastern Europe. The Mediterranean basin (6°W, 36E,30°N, 46°N) is affected both by anthropogenic, biogenic and dust
emissions (Dayan et al., 2020; Chazette et al., 2019; Michoud et al., 2017; Pace et al., 2006). Its AOD seasonal cycle shown
in Figure 3 and ranging between 0.07−0.19 (average values obtained from the ocean part of the basin), follows the Saharan
dust transport cycle for the southern part, whereas the northern part is mostly dominated by human activities. In fact, the high
AOD MAM values (between 0.2 and 0.3) shown over the South–South East part of the Mediterranean basin are caused by the
on–set of the Saharan dust transport due to a low–pressure system (the Sharav cyclone), which pushes the dust plumes to the
eastward basin (Floutsi et al., 2016; Moulin et al., 1998). During summer, the Balearic cyclogenesis is causing the spreading
of the dust plumes northwards from the Saharan source region, explaining the high JJA values (AOD>0.2) over large areas of
the Southern part of the basin (Formenti et al., 2018; Floutsi et al., 2016; Moulin et al., 1998). The AOD average over February
2000−August 2021 period over the Mediterranean basin (6°W, 36E,30°N, 46°N) resulted in an AOD of 0.13 at 550 nm,
comparable to the result obtained in (Chiapello et al., 2021) at 865 nm based on POLDER–3 observations.

In the Po Valley (7°E, 12°30'E, 43°36'N, 46°12'N), the seasonal cycle is ranging between 0.09 (for DJF) to 0.15 (for JJA)
with maxima in June, July (AOD>0.16). Particulate matter (PM) composition measurements at the ground show to be
dominated by traffic, biomass burning emissions, as well as ammonium nitrate and sulfate formation (Scotto et al., 2021) and
the largest ground PM values can occur in DJF and SON seasons due to recurrent low temperatures and possible intense
residential biomass burning (Pietrogrande et al., 2015) and ammonium nitrate precursor emissions (Scotto et al., 2021; Vecchi
et al., 2018). The MAM and JJA levels at the ground can be caused by agricultural local sources (e.g. burning of pruning and
fertilizers) (Scotto et al., 2021; Bucci et al., 2018; Clarisse et al., 2009). In summertime high levels of secondary organic
aerosols in presence of stagnation conditions have been also observed in (Sandrini et al., 2016). Since AOD values are
columnar values, the difference observed in the seasonal cycle between the ground (mainly DJF–MAM peaks) and AOD (JJA)
can be attributed to different reasons: 1) the planetary boundary layer height (PBL), lower in the winter and higher in the
summer, which conversely to ground PM is not affecting the AOD measurements and 2) possible dust events and biomass
burning fires that can contribute to the stronger AOD levels during the spring and summer seasons. In addition, high AOD
levels are also favored by insufficient pollution dispersion and removal, the valley being surrounded by mountains (the Alps



and the Apennines), especially under stable weather conditions, promoting pollutants accumulation and air masses stagnation
(Putaud et al., 2014). As a matter of fact, this reasoning is general and not restricted to Po valley.

For what concerns Eastern Europe (13°E, 30°E, 42°N, 55°N), Figure 3 shows a strong seasonal variability of AOD for regions
like Poland and Serbia, with maximum AOD of up to 0.2 during the JJA season and minimum over the DJF season with AOD
generally below 0.15. Seasonal cycle with a maximum over summer and spring has been also observed in (Chubarova et al.,
2016) for the 2001–2014, studying the Moscow AERONET site. Furthermore, (Bovchaliuk et al., 2013), found AOD values
ranging between 0.05 and 0.2 at 870 nm for the 2003–2011 period, with peaks over the spring, and which the authors explain
by agricultural fires correlated with an observed increase in the fine fraction mode particles.

Finally, also the Western Europe (11°W, 18°E, 35°N, 60°N) shows an AOD seasonal variability over land in the [0.06–0.12]
range with the maximum on the JJA season. A summer AOD maximum attributable to dust, smoke and sea–spray aerosols has
been also found in (Zhao et al., 2018) for this area. These values are lower compared to other regions within Europe. This
result can be justified by its proximity to the Atlantic Ocean, which contribute to expose these areas to more humid and less
polluted air masses as well as to a greater pollutants dispersion capability.

### 3.2 City scale analysis

The 1 km resolution MAIAC AOD data are used to explore AOD levels over cities and evaluating and quantifying the extra
AOD of cities with respect to their surrounding areas. Figure 4 shows the distribution and the heterogeneity of the aerosol
optical depth over the different sites. European cities are ordered by increasing 50th percentile values of the local scale AOD
from left to right. Table 1 gives the coordinates of city centers. An example of AOD time series for some of the cities is shown
in Figure S3. In line with Section 3.1, a North–South gradient can be found among the cities as well, highlighting that European
cities located at more northerly latitudes have AOD levels at 550 nm in general lower compared to cities at more southerly
latitudes: Oslo, Dublin and London, show AOD median values (the 25th and 75th percentiles are also reported in brackets) of
0.06 [0.03−0.10], 0.06 [0.04−0.10] and 0.07 [0.04−0.12] respectively, whereas more southerly located Bologna, Milan and
Athens show 0.13 [0.08−0.19], 0.14 [0.07−0.23], 0.16 [0.11−0.23] respectively. Sites located in the middle range of the Figure
4, like Lisbon, Berlin and Amsterdam, show AOD median values 0.09 [0.07−0.12], 0.10 [0.05−0.16], 0.09 [0.06−0.16]. The
AOD values in the northern cities are only rarely exceeding a threshold of 0.3, which we arbitrarily relate to pollution
(anthropogenic, dust, fires…) events (0.6%, 2.5% and 4.0% of the total observations for Oslo, Dublin and London respectively,
see Table 2). For more southerly cities like Milan and Athens, this fraction is 14.7% and 10.6% respectively. Looking more
closely to the timing of these occurrences, 18% and 10% of these "high pollution" cases occurred before and after 2010
respectively for Milan, and 14% and 8% for Athens.

Figure 4 shows that the city center local scale AOD levels are most of the time larger than the regional AOD. As well, an
increase in the frequency of AOD>0.3 can be also observed for the city AOD (See Table 2). The local–to–regional ratio
(LTRR) calculated using Eq. (1) for the 2000–2021 period and for the different cities is summarized in Table 2. Again, positive
LTRR values are characteristic for an urban scale contribution to the aerosol burden on top of the regional one, highlighting
the importance of local anthropogenic emissions and atmospheric conditions favorable to pollutants accumulation. For
instance, a LTRR value of 1 would correspond to a 50% contribution of local urban aerosol to total AOD, while a value of 0.5
to a contribution of a third. It should be noted that the local contribution to surface PM is necessarily stronger than that to
AOD, as the importance of the regional background is more important for the vertical column.

On the contrary, negative LTRR values indicate a lower local city AOD than the regional one, suggesting a local aerosol loss
at urban scale. However, systematic urban loss processes are not easy to identify. Sedimentation and dry deposition processes



are not expected to be particularly enhanced over urban areas, nor is precipitation, compared to its regional surrounding. On the other hand, the urban heat island with increased temperatures could lead to evaporation of particles. For instance, (Pirhadi et al., 2020) finds that due to its semi-volatile character, about 50% of ambient PM2.5 aerosol evaporated when heated up in a thermo−denuder from ambient temperature (~13°C in winter, 23°C in summer and up to 50°C). The urban heat island effect depends on the size and additional heat production within an urban area. It is restricted to light wind meteorological conditions and it is more pronounced during night, while MAIAC observations are made during daytime. For these reasons, we consider that evaporation of semi–volatile aerosol under higher urban temperatures could only play a limited effect in our dataset.

Since the observed negative LTRR values were in general very small (in the order of some %) an alternative explanation to negative LTRR values is a possible inhomogeneity in AOD within the rather large (100 x 100 km$^2$) regional domain. This could be true especially for coastal sites, or partly mountainous sites. In the frame of the present analysis it is in general difficult to distinguish between these two loss and inhomogeneity effects.

Maximum mean values of LTRR are shown for Barcelona (0.57 ± 0.02), Lisbon (0.55 ± 0.03), Paris (0.39 ± 0.02) and Athens (0.32 ± 0.01). On the contrary, significantly negative LTRR values are shown for Brussels (–0.06 ± 0.01), Amsterdam (–0.17 ± 0.01), Berlin (–0.03 ± 0.01). The uncertainty has been calculated here as the standard error of the mean: $\sigma/\sqrt{N}$, where $\sigma$ is the standard deviation of the LTRR distribution, and N is the number of points available over the 2000–2021 period. The most negative LTRR is found for Amsterdam. For this coastal city, larger AODs are observed over the sea than over the continent (see Figure 3 especially for the spring and autumn seasons) which could be caused by enhanced sea–salt, but possibly also by slight differences in the retrieval algorithm for sea and land surfaces. Thus, the regional background cannot be considered as homogeneous for this case.

In this study we focus on the Paris area which shows a LTRR of 0.39 ± 0.02. The aim of this interest is supporting the preceding climatological studies performed for the ACROSS field campaign (Cantrell and Michoud, 2022). Paris represents a strongly centralized agglomeration with about 11 million inhabitants. It is located in Western Europe, in a rural area without strong orography, and at some 200 km from the Atlantic Ocean. This leads to generally favorable pollutant dispersion conditions (Vautard et al., 2003). The median local AOD value at 550 nm is 0.10 [0.07−0.15] for Paris which falls slightly over the median of the cities distribution in Figure 4. Results from MEGAPOLI (Beekmann et al., 2010) campaign have shown that large fraction of fine PM at the ground is transported from the European continent and southern France towards Paris, while local emissions represent a smaller fraction (Beekmann et al., 2015; Bressi et al., 2014). Later studies with multi–year data sets (mid 2011– mid 2013, (Petit et al., 2015)) or pointing to specific pollution episodes (December 2016, (Foret et al., 2022)) make evident the local emission contribution to fine aerosol pollution peaks. (Skyllakou et al., 2014) shows by source apportionment that primary organic aerosol (POA) and elemental carbon (BC) are controlling the PM2.5 fraction of the Paris local emissions, whereas regional advection is controlling the secondary PM2.5 fraction. Organic aerosols have been shown to play a key role in the Paris air quality assessment (Zhang et al., 2019; Petit et al., 2015; Bressi et al., 2014). Sulfate and secondary organic aerosols are mainly attributed to long–range transport (Foret et al., 2022; Skyllakou et al., 2014). For the period 2000–2021, the % of days with AOD>0.3 is found to be 4% at the Paris local scale.

Barcelona shows a local/regional distribution rather similar to Paris (Figure 4) although its LTRR is larger (0.57 ± 0.02, actually the largest in our study) and the geomorphology of the two sites is significantly different. Barcelona shows the largest LTRR (0.57 ± 0.02) among the cities studied. It is located on the northeast part of the Iberian Peninsula, bordering the Mediterranean Sea and the foot hills of the Pyrenees mountains. Re–circulation caused by mountain winds and sea breeze (Jaén et al., 2021; Pérez et al., 2004) could enhance the local urban AOD. In addition, regional background over the mountainous area next to



Barcelona is relatively low with respect to that over the Mediterranean see (Figure 3), which could contribute to the large LTRR.

The Bologna and Milan surrounding is the well–known Po–Valley, as previously discussed, where recirculation and stagnation
events of aerosol and precursors may occur and cause enhanced pollution levels (Putaud et al., 2014). (Vecchi et al., 2018) showed by source apportionment analysis that, during the winter season, major PM contribution to light extinction for the Milan urban area is nitrate (42%), followed by sulfate, primary aerosol due to traffic and biomass burning related organic aerosol. In another study, secondary inorganic aerosols have been also shown to contribute with 35% on PM over the Milan urban area on the annual average (Amato et al., 2016). As a consequence, the large regional PM background leads to
comparatively small additional local contributions and small LTRR values for these both cities.

For what concern Athens with a LTRR of $0.32\pm 0.01$, aerosols of anthropogenic–origin have been shown to dominate. Indeed (Taghvaee et al., 2019) showed by source apportionment analysis that traffic emissions, SOA and biomass burning correspond to major sources to PM2.5 samples, contributing respectively 44%, 16% and 9%, with higher PM values during summer than winter. During the latter season high PM2.5 episodes are linked to dust and biomass burning episodes (Raptis et al., 2020).
Furthermore, the organic aerosol concentrations in Athens have been shown to be dominated by regional SOA during summertime (Tsiflikiotou et al., 2019), highlighting also the importance of long–range transport in this area (Manousakas et al., 2020). As a conclusion of this discussion, PM2.5 sources over the Athens region are a mixture of regional and local origin, which is reflected its LTRR value.

For the seasonal variation of LTRR values, Figure 5 shows the scatter plot between the local and regional AOD as a function
of the season for all cities. The fitting line considering all seasons shows a slope of 1.08 and a linear correlation of 0.83, highlighting the overall average positive contribution on air quality degradation of the local on the regional scale over the different seasons. Figure 5 shows that the fraction of points where $AOD_{local}>AOD_{regional}$ is 84 %, 65 %, 75 %, 97 % for DJF, MAM, JJA, SON respectively. This result suggests that the local contribution is higher during winter and lower during summer. In order to explain this difference, it should be considered that during summer time, favorable weather condition, stronger
photochemistry activity and enhanced BVOC emissions can lead to increased secondary aerosol formation, and increase the secondary to primary aerosol ratio. As secondary aerosol formation is a regional phenomenon (Beekmann et al., 2015; Skyllakou et al., 2014; Karl et al., 2009), the regional contribution to AOD is increased. Furthermore, possible dust and fire events can also contribute to the increase of the regional signal during summer and spring over Europe. However, during wintertime secondary aerosol formation is less pronounced, in addition primary aerosol emissions are increased due increased
heating demand.

### 3.3 Trend Analysis

The analysis of the high resolution MAIAC product can contribute to further investigate the aerosol optical depth tendency over the European region. Statistically significant (pvalue<0.05) absolute and relative AOD trends over the European continent are reported in Figure 6. Negative AOD trends have been found over the domain of interest, in the [–3; –0.6] %year[-1] range,
representing the 5[th] and 95[th] percentile respectively of the Figure 6b. Furthermore, more negative trends are mostly found over the regional hotspots outlined in section 3.1 (Po valley, Mediterranean basin, parts of eastern Europe). Decreasing relative and absolute trends of $–1.34 \pm 0.29$ %year[-1] and $–0.0021\pm0.0005$ units year[-1] for the Mediterranean Basin has been found for the 2001–2021 period. A decreasing absolute trend of –0.003 units year[-1] for the 2002–2014 period has been also found with the MAIAC data in agreement to the –0.003 units year[-1] observed in (Floutsi et al., 2016). A trend of $–1.66 \pm 0.58$ %year[-1] at
550nm has been estimated for the Po Valley, lower than what has been observed at the Ispra AERONET site in the period



2004–2010 (Putaud et al., 2014) where they estimated a decreasing trend of −4.0 ± 1.8 and −2.5 ± 1.3 %year[-1] for the 440nm and 675nm respectively. Negative AOD tendency has been also registered for the Benelux and the Peloponnese area of –2.46 ± 0.96 %year[-1] and –1.49 ± 0.45 %year[-1] respectively. A statistically absolute significant trend of –0.003 ± 0.002 units year[-1] has been observed for the Eastern Europe area, in line to what observed in (Filonchyk et al., 2020a) for the 2002–2018 period,

where values in the range of [–0.0025;–0.0028] units year[-1] are observed for Czech Republic, Bulgaria, Slovakia and Hungary with MODIS TERRA data. However stronger trends for the 2002–2019 period, in the range of [–0.0031/–0.0076] units year[-1], are observed in (Filonchyk et al., 2020b) for several cities in the Eastern Europe, by using MODIS AQUA data. (Chubarova et al., 2016) attributes the significant negative trends observed in Moscow for the 2001–2014 period to the strong decrease in $SO_x$, non–methane volatile organic compounds (NMVOC) and $NO_x$ emissions. As a matter of fact, (Tsyro et al., 2022)

predicted, through a six–models ensemble approach, a decreasing of surface PM2.5 and PM10 over Europe for the 1990–2010 period, attributing a large impact to sulfate, ammonium and nitrate precursor emission reductions. Nevertheless, this decrease appeared to be more impacting over Central and Eastern Europe. For instance, trends stronger than –2.5 %year[-1] are observed over Germany, Czech Republic, Hungary, Slovakia and Ukraine both for PM2.5 and PM10.

Taking advantage of the high spatial resolution of the MAIAC product, the analysis has been extended to a local scale to

estimate the AOD trends over the cities listed in the Table 2 and to compare them to the trends for the surrounding regional background. Results are shown and summarized in Table 3. For most of the sites, a significant negative trend can be identified consistently both within the city center (3x3km[2]) and in the surrounding area (100x100km[2]). For instance, Athens, Prague, Vienna, Milan, Zagreb and Bologna show AOD trends of in the range of [–0.9; –1.7] %year[-1] and [–1.3; –2.0] %year[-1] for the regional and local scale respectively. This result is in line with the aforementioned observations at the European scale and with

other studies focusing on European megacities (Papachristopoulou et al., 2022; Gupta et al., 2022; Zhao et al., 2017). (Papachristopoulou et al., 2022) observed a decrease in AOD up to –0.03 units decade[-1] over the 2003–2020 period and for European megacities like Paris, Barcelona, Madrid and London. This result is comparable to the range of [–0.01; –0.03] units decade[-1] observed for the European cities analyzed in this study. Conversely, cities where AOD levels are relatively low (positioned in the leftmost part of the Figure 4) do generally not show statistically significant results. Among all the cities,

Prague shows the strongest relative trend at both regional and local scales of –2.0 %year[-1] and –1.7 %year[-1] respectively. The absolute value obtained in this study for the local scale is comparable to −0.0022 obtained in (Filonchyk et al., 2020a) for the 2000–2018 period. Moreover, the –1.0 %year[-1] estimated trend over Athens is in line with the –1.1 %year[-1] obtained at 440 nm at the AERONET urban site in (Raptis et al., 2020) for the 2000–2018 period. In the case of Paris, a trend of –1.5 %year[-1] is obtained for the regional scale, while the city center trend is not significant. Interestingly, the regional relative trend is for

most cities (9 of 11 in Table 3) stronger (i.e. more negative) than the local one. One possible reason of this outcome could be a stronger decrease of secondary aerosols due to stringent pollution control of precursors (SO2, NOx, VOC) than that of primary aerosol, as found for several French EMEP/MERA network sites (Font et al., 2023). Indeed, the primary to secondary aerosol ratio is expected to be larger for urban than for regional background sites.

## 4. Conclusions

This study presents a quantitative estimation of the aerosol optical depth variability in Europe using long–term measurements (2000–2021) from the MAIAC algorithm applied to MODIS satellite observations. The MAIAC validation, performed at the European scale against ground–based sun photometer data, demonstrates a slight underestimation of MAIAC AOD, showing a MBE of –0.02 and a RMSE of 0.06 respectively. An expected error $EE=\pm(0.05AOD + 0.05)$ has been found for the European continent, lower with respect to what suggested by (Lyapustin et al., 2018). Moreover, according to the AERONET

AE splitting analysis, the validation, which provides satisfactory results for mixed− and fine− aerosol mode, is less performant



in presence of coarse–dominated aerosols. This suggests that further improvements of the MAIAC algorithm are needed for scenes dominated by dust or other coarse–sized particles.

Regarding the AOD seasonal climatology over the European continent, the AOD exhibits maximum and minimum values during the summer (JJA) and winter (DJF) seasons, respectively, showing a strong North–South latitudinal gradient. Values

of AOD in the range of 0.12−0.22 (JJA) and 0.06−0.09 (DJF) are observed in the 30°N−60°N band.

Concerning the link between regional and local scales air quality across the main European cities that was the main objective of this work both the regional background and city level AOD show a general north–south gradient with increasing AOD and several hotspots over the Po valley and the Mediterranean Sea. The local–to–regional analysis shows that most of the cities contribute to enhance the AOD loading with respect to their regional background. On the contrary, for some cities a slightly

negative LTRR could be either explained by specific losses or by an inhomogeneity of the regional background. On a relative scale the city contribution to regional AOD is maximum during the winter season, because the primary vs. secondary aerosol ratio is expected to be the largest. Concerning the Paris area, most of pollution has been considered as transported from the European continent in previous studies (Beekmann et al., 2015). In fact, Paris represents an important isolated agglomeration with respect to the surrounding area. Indeed, the long–term analysis conducted in this work indicates an average local–to–

regional ratio of 39%, suggesting a non–negligible impact of the city emissions in addition to the regional aerosol burden in Paris. Further investigation is needed to understand the nature of this AOD difference. As a matter of fact, the interaction between the regional background and the local emissions cannot be exploited through AOD measurements directly, although we know that changes in the chemical and optical properties lead to changes in the aerosol extinction profile. Further investigation on the interaction between biogenic and anthropogenic local and regional air masses and the impact on aerosol

properties will be provided in the ACROSS project (Cantrell and Michoud, 2022).

Different studies have already shown a negative decreasing AOD trends over the European continent using in particular MODIS satellite data. However, most of the time with a broader spatial resolution with respect to the product used in this study. The MAIAC high spatial resolution product has been exploited to investigate the AOD trends both at European and local city scale. The result showed a general AOD decrease over the all European continent, consistent with the recent literature

and in connection with the mitigation policies over the European countries. In particular, spatial homogeneous trends have been found over known European hotspots (e.g. –1.34 %year$^{-1}$, –1.66%year$^{-1}$, for the Mediterranean Sea, the Po Valley respectively). In addition, taking advantage of the high spatial resolution, the analysis has been extended also at the city level, showing a statistically significant yearly decrease during the last two decades in AOD at 550 nm in the range of [–0.5; –1.7] %year$^{-1}$ and at city level and [–0.9; –2.0] %year$^{-1}$ in the surroundings. This result highlights the faster decrease in regional

AOD levels with respect to those at the urban local scale. Nevertheless, over the Paris area, we observed a statistically significant negative trend only at the regional scale. A potential explanation could be linked to the more stringent control of aerosol precursors emissions (SO$_2$, NOx, VOC) with respect to direct aerosol emissions (Font et al., 2023).

**Author contributions**

LDA, CDB, and MB designed the study and discussed the results. LDA performed the data analysis with contributions by

CDB and MB. GF, PF, GS, and JFD contributed to the discussion of the results. LDA, CDB, and MB wrote the paper with contribution from all co–authors.



**Data access**

The MAIAC MCD19A2 data used in this study are accessible at: https://lpdaac.usgs.gov/products/mcd19a2v006/ (last access: June 2023). The AERONET data are available at https://aeronet.gsfc.nasa.gov/ (last access: June 2023). Population data

reported in Table 1 can be accessed at the following link: https://ec.europa.eu/eurostat/cache/RCI/#?vis=city.statistics&lang=en (last access: June 2023). The data and the codes that support the findings of this study are available upon request from the corresponding authors.

**Competing interests:**

The authors declare that they have no conflict of interest.

**Acknowledgements**

This work has been supported by the ACROSS and the RI–URBANS projects. The ACROSS project has received funding from the French National Research Agency (ANR) under the investments programme integrated into France 2030, with the reference ANR–17–MPGA–0002, and it was supported by the French National program LEFE (Les Enveloppes Fluides et l'Environnement). The RI–URBANS project has received funding from the European Union's Horizon 2020 research and

innovation program under grant agreement No 101036245. We thank all the AERONET PIs and their staff for establishing and maintaining all the sites used in this investigation. Useful discussions with M. Mallet, Y. Derimian, and J. C. Raut are gratefully acknowledged. We thank C. Cantrell and V. Michoud, PIs of the ACROSS project.

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





**Figures**

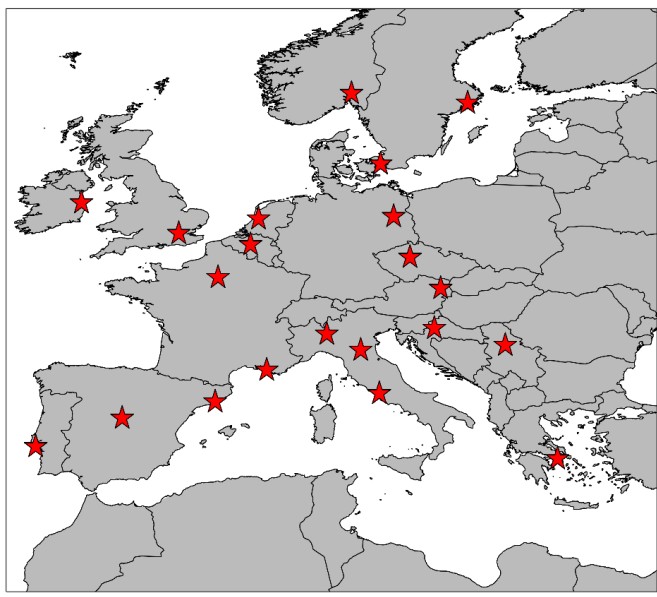


**Figure 1: Localization of European cities used for the local–to–regional analysis. Map created with Cartopy (Met Office, 2010).**




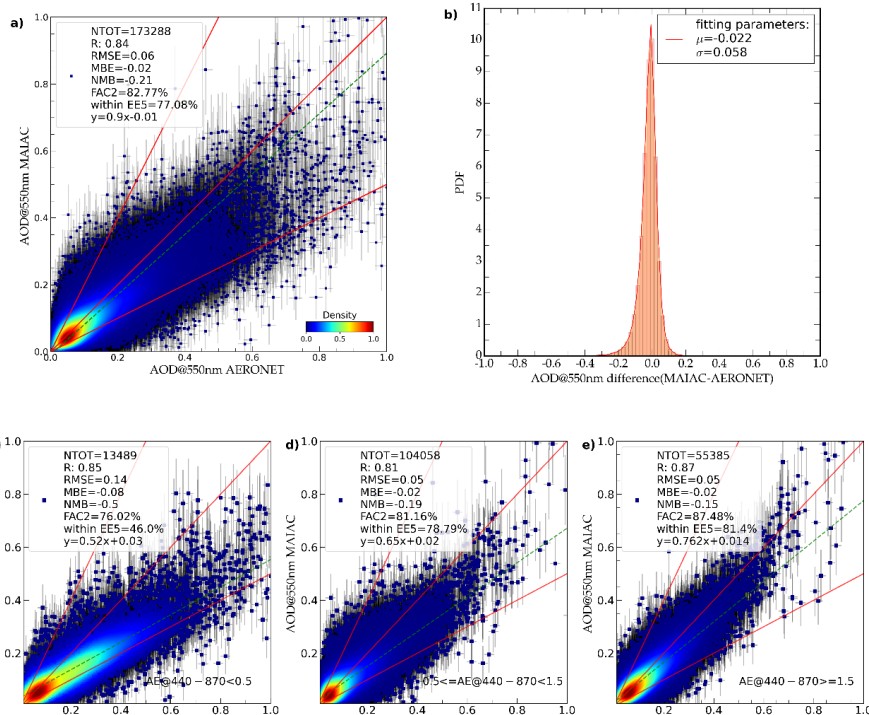


**Figure 2: Scatterplot of the MAIAC against AERONET observations considering a) all the available data points in Europe, and points selected based on the Angstrom Exponent (AE) assuming c) AE<0.5 d) 0.5≤AE<1.5 e) AE≥1.5. Panel b) shows the PDF of the difference between MAIAC and AERONET values in reference to data points in a). Acronyms indicate: total number of points (NTOT), correlation coefficient (R), Root Mean Square Error (RMSE), Mean Bias**
**Error (MBE), Normalized Mean Bias (NMB), fraction of y data between 0.5 and 2 times x (FAC2), fraction of retrievals within the expected error (EE5=0.05+0.05AOD) and the equation of the regression line. Vertical and horizontal bars represent the x and y errors.**





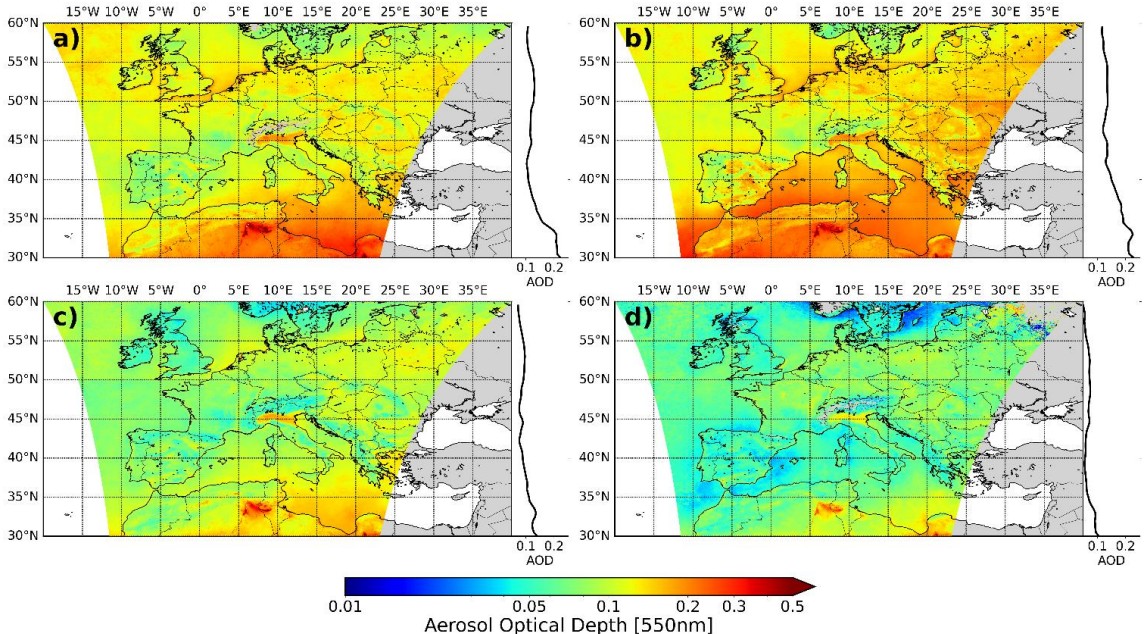


**Figure 3: Climatological seasonal mean of the Aerosol Optical Depth at 550 nm from MAIAC algorithm over the period 2000–2021. Seasons are ordered as follows: (a) March–April–May, (b) June–July–August, (c) September–October–November, and (d) December–January–February. The right side of each figure shows the latitudinal average of AOD.**




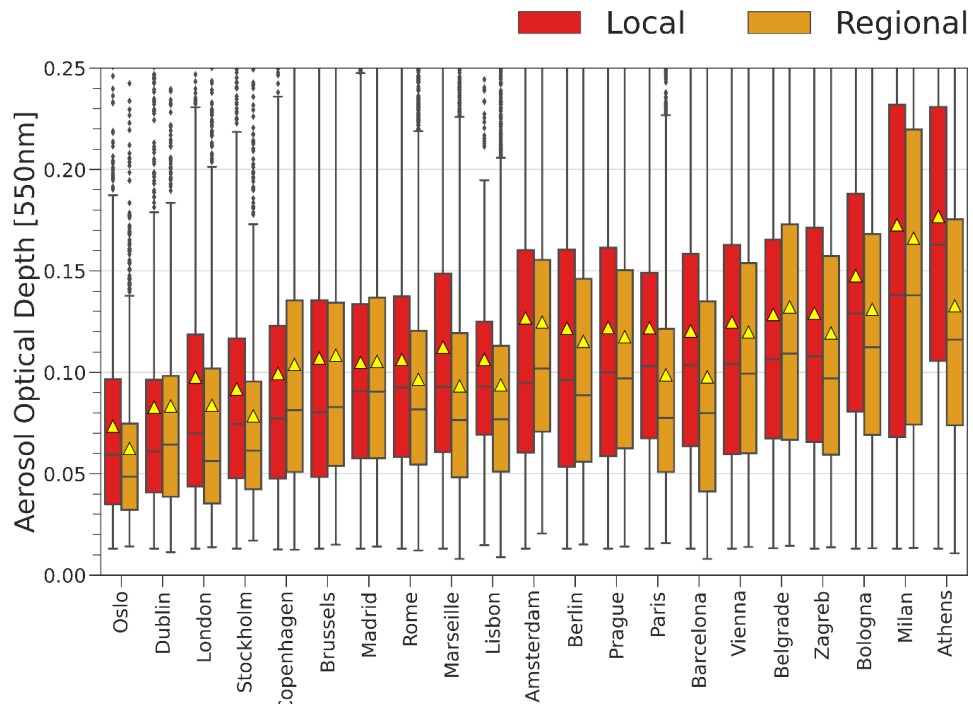

**Figure 4: Climatology of the Aerosol Optical Depth at 550nm from MAIAC algorithm at different cities in Europe for (red) the local scale, and (orange) the regional scale, as defined in Sect. 2.1. The location of the cities is shown in Figure 1. The figure aims to enhance the contribution to the AOD enhancing due to the local source of pollution. The yellow triangles represent the mean of the boxplot, whereas the median has been reported as the line crossing the boxplot. Black dots represent the outliers.**



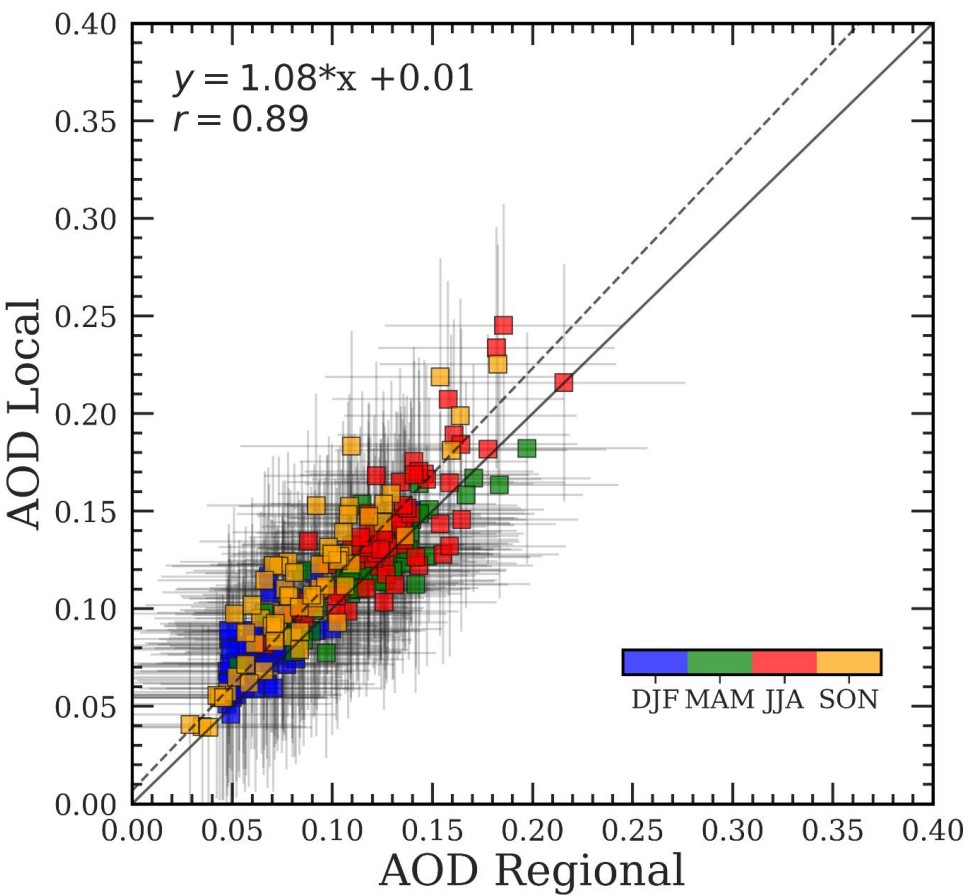

**Figure 5: Scatter plot Local Vs Regional at different seasons (December–January–February (DJF), March–April–May (MAM), June–July–August (JJA), September–October–November (SON)). During the DJF and MAM seasons, the largest differences between local and regional are found. Vertical and horizontal bars represent the x and y errors.**







**Figure 6: Theil–Sen (a) absolute and (b) relative change of Aerosol Optical Depth at 550 nm over the European domain for the 2000–2021 period. Only the significant (pvalue<0.05) pixels are reported.**




**Tables**

| LONGITUDE | LATITUDE | CITY |
|---|---|---|
| 4.88 | 52.37 | Amsterdam |
| 23.72 | 37.98 | Athens |
| 2.15 | 41.39 | **Barcelona** |
| 20.43 | 44.80 | **Belgrade** |
| 13.40 | 52.52 | **Berlin** |
| 11.32 | 44.49 | Bologna |
| 4.38 | 50.83 | **Brussels** |
| 12.57 | 55.68 | Copenhagen |
| −6.26 | 53.349 | Dublin |
| −9.13 | 38.72 | Lisbon |
| −0.12 | 51.50 | **London** |
| −3.70 | 40.41 | **Madrid** |
| 5.4 | 43.3 | Marseille |
| 9.18 | 45.46 | **Milan** |
| 10.75 | 59.91 | Oslo |
| 2.33 | 48.86 | **Paris** |
| 14.43 | 50.07 | **Prague** |
| 12.49 | 41.90 | **Rome** |
| 18.06 | 59.33 | Stockholm |
| 16.36 | 48.21 | **Vienne** |
| 15.97 | 45.81 | Zagreb |


**Table 1: List of European cities used for the city scale analysis. City names in bold are the cities with more than 1 million of inhabitants according to the Eurostat database (https://ec.europa.eu/eurostat/cache/RCI/#?vis=city.statistics&lang=en).**






| N DAYS | AOD>0.3 (%) | LOWER/UPPER BOUND [25th/75th] | AOD>0.3 (%) | LOWER/UPPER BOUND [25th/75th] | MEAN ± STD | LOWER/UPPER BOUND [25th/75th] | |
|---|---|---|---|---|---|---|---|
| | AOD LOCAL | | AOD REGIONAL | | LTRR | | CITY |
| 886 | 5.3 | 0.06/0.16 | 4.1 | 0.07/0.16 | −0.17 ± 0.01 | −0.31/−0.04 | Amsterdam |
| 4081 | 10.6 | 0.11/0.23 | 3.2 | 0.07/0.18 | 0.32 ± 0.01 | −0.04/0.52 | Athens |
| 3121 | 3.5 | 0.06/0.16 | 1.6 | 0.04/0.13 | 0.57 ± 0.02 | 0.02/0.93 | Barcelona |
| 2424 | 5.2 | 0.07/0.17 | 5.9 | 0.07/0.17 | 0.07 ± 0.01 | −0.15/0.24 | Belgrade |
| 1631 | 5.3 | 0.05/0.16 | 4.1 | 0.06/0.15 | −0.03 ± 0.01 | −0.17/0.11 | Berlin |
| 3222 | 7.1 | 0.08/0.19 | 4.8 | 0.07/0.17 | 0.14 ± 0.01 | −0.07/0.28 | Bologna |
| 1389 | 4.2 | 0.05/0.14 | 4.1 | 0.05/0.13 | −0.06 ± 0.01 | −0.21/0.07 | Brussels |
| 1037 | 3.1 | 0.05/0.12 | 2.4 | 0.05/0.14 | −0.01 ± 0.02 | −0.25/0.18 | Copenhagen |
| 910 | 2.5 | 0.04/0.10 | 2.6 | 0.04/0.10 | −0.01 ± 0.02 | −0.32/0.16 | Dublin |
| 445 | 1.3 | 0.07/0.12 | 1.5 | 0.05/0.11 | 0.55 ± 0.03 | 0.15/0.88 | Lisbon |
| 1080 | 4.0 | 0.04/0.12 | 3.1 | 0.04/0.10 | 0.13 ± 0.02 | −0.08/0.29 | London |
| 3049 | 1.5 | 0.06/0.13 | 1.6 | 0.06/0.14 | 0.14 ± 0.01 | −0.04/0.29 | Madrid |
| 4394 | 2.4 | 0.06/0.15 | 1.2 | 0.05/0.12 | 0.26 ± 0.01 | −0.13/0.48 | Marseille |
| 3220 | 14.7 | 0.07/0.23 | 12.5 | 0.07/0.22 | −0.01 ± 0.01 | −0.18/0.15 | Milan |
| 1042 | 0.6 | 0.03/0.10 | 0.6 | 0.03/0.07 | 0.07 ± 0.02 | −0.11/0.19 | Oslo |
| 1293 | 4.4 | 0.07/0.15 | 2.5 | 0.05/0.12 | 0.39 ± 0.02 | 0.01/0.64 | Paris |
| 1502 | 4.5 | 0.06/0.16 | 3.6 | 0.06/0.15 | −0.03 ± 0.01 | −0.21/0.14 | Prague |
| 3670 | 1.9 | 0.06/0.14 | 1.2 | 0.05/0.12 | 0.10 ± 0.01 | −0.13/0.27 | Rome |
| 1119 | 1.8 | 0.05/0.12 | 1.8 | 0.04/0.10 | 0.04 ± 0.02 | −0.10/0.15 | Stockholm |
| 2015 | 4.9 | 0.06/0.16 | 3.5 | 0.06/0.15 | 0.03 ± 0.01 | −0.19/0.2 | Vienna |
| 2531 | 5.2 | 0.07/0.17 | 3.8 | 0.06/0.16 | 0.08 ± 0.01 | −0.10/0.24 | Zagreb |

**Table 2: Aerosol Optical Depth statistics at 550 nm from MAIAC algorithm at different sites both for local and regional scale: number of days AOD>0.3, 25th and 75th distribution percentiles, AOD local–to–regional ratio mean ± its standard deviation, and AOD local–to–regional ratio 25th and 75th distribution percentiles are reported.**






| TREND | | | | |
|---|---|---|---|---|
| **AOD LOCAL** | | **AOD REGIONAL** | | |
| **ABSOLUTE** (units year⁻¹) | **RELATIVE** (% year⁻¹) | **ABSOLUTE** (units year⁻¹) | **RELATIVE** (% year⁻¹) | **CITY** |
| – | – | – | – | Amsterdam |
| −0.0017 | −1.0 | −0.0020 | −1.3 | Athens |
| −0.0010 | −0.9 | −0.0017 | −1.5 | Barcelona |
| −0.0016 | −1.6 | −0.0014 | −1.2 | Belgrade |
| −0.0015 | −1.4 | – | – | Berlin |
| −0.0021 | −1.4 | −0.0025 | −1.8 | Bologna |
| −0.0020 | −1.5 | −0.021 | −1.5 | Brussels |
| – | – | – | – | Copenhagen |
| – | – | – | – | Dublin |
| – | – | – | – | Lisbon |
| – | – | – | – | London |
| – | – | – | – | Madrid |
| −0.0005 | −0.5 | −0.0014 | −0.9 | Marseille |
| −0.0034 | −1.4 | −0.0033 | −1.7 | Milan |
| – | – | – | – | Oslo |
| – | – | −0.0015 | −1.5 | Paris |
| −0.0030 | −1.7 | −0.0030 | −2.0 | Prague |
| −0.0012 | −1.1 | −0.0014 | −1.3 | Rome |
| – | – | – | – | Stockholm |
| −0.0011 | −0.9 | −0.0025 | −1.9 | Vienna |
| −0.0020 | −1.6 | −0.0022 | −1.8 | Zagreb |

**Table 3: Optical Depth trends at local and regional scale for the different analyzed cities. Only significant trends are shown (pvalue<0.05).**