# Peer review of "Aerosol optical depth climatology from the high-resolution MAIAC product over Europe: differences between major European cities and their surrounding environments"

_EGUsphere, 2023_

## Author Comment (AC1)

At first, we would like to thank the reviewers for having carefully read the paper and provided valuable comments which helped to improve the quality of the manuscript. We have taken into consideration all the comments raised by the reviewers, and changed the paper accordingly. The details of our changes are highlighted in the text. The point by point answers to Reviewer #1 and #2 are provided in the following.

**Reviewer #1**

**Specific comments**

Lines 132-133: This minimum limit is for both local and regional scale, right?
Yes, the limit is for both. This aspect has been clarified in the main manuscript, line 135.

Lines 132-135: Before this calculation, which type of aggregation was performed to the daily values of the two scales considered here? Inside the local and inside the regional scale?
Both for the local and the regional scale the mean of all the pixel inside the 3x3 km$^2$ and 100x100 km$^2$ boxes has been performed with a threshold of 40% on the available data. The LTRR ratio has been calculated afterwards. This aspect has been clarified in the main manuscript, lines 136-137.

Lines 149-153: To my opinion the MAIAC AOD retrievals uncertainty could be discuss better when you first introduce the MAIAC product in the first paragraph of the 2.1 Section.
The following sentence "The uncertainty attributed to the MAIAC AOD retrievals has been defined through the expected error (EE) considering both absolute and relative errors by attributing an absolute error of 0.05 and a relative error of 0.1 following (Falah et al., 2021; Lyapustin et al., 2018). As discussed in the next Section, the validation against AERONET will be considered to reevaluate the EE over Europe and subsequently update the MAIAC uncertainty."
has been moved to Line 114 and elaborated as follows:
"The uncertainty attributed to the MAIAC AOD retrievals has been defined through the expected error EE=$\pm(0.1 AOD_{AERONET} + 0.05)$, indicating the percentage of $AOD_{MAIAC}$ retrievals falling in the envelope (expressed in %). The EE has been established following (Falah et al., 2021) and (Lyapustin et al., 2018), considering both absolute and relative errors and by attributing an absolute error of 0.05 and a relative error of 0.1. As discussed in the next Section, the validation against AERONET will be considered to reevaluate the EE over Europe and subsequently update the MAIAC uncertainty."

Lines 160: Please check again formula 2, in the numerator you mean 550nm?
Yes, the reviewer is right and the formula has been corrected accordingly. Thank you for this correction.

Line 165: I don't follow here why the area from the MAIAC product is expressed in deg x deg and not only in km x km? As has already been mentioned in the previous section.
In order to extract the data for the MAIAC validation against the AERONET network we used a box of 0.06°x0.06°. Hence, the area has been reported in degree to be consistent with the method used. In addition, the sentence "In order to improve the meaningfulness against the

AERONET observations, the MAIAC AOD have been additionally extracted by taking an average area of 0.06°x 0.06° over the AERONET site, corresponding to ~7x7 km2. Indeed (Falah et al., 2021) show that MAIAC – AERONET comparisons given similar results for boxes between 1x1 km2 and 9x9 km2.", has been elaborated as follows: "In order to improve the meaningfulness against the AERONET observations, the MAIAC AOD have been additionally extracted by taking an arbitrary average area of 0.06°x 0.06° over the AERONET site, corresponding to ~7x7 km2, chosen between the 1x1 km2 and 9x9 km2 boxes for which (Falah et al., 2021) show that MAIAC – AERONET comparisons give similar results."

Line 186: Could you please provide this equation before this sentence, elaborating a little how it is extracted?
The expected error equation has been further described as indicated in a previous comment (see Lines 149-153 comment).

Lines 199 & 201: The "EE is …%"? could you please rephrase according to lines 189-199?
The sentences "However, for the coarse−mode the MBE (–0.08) and EE (46%) are significantly lower with respect to the other two classes. In case of AOD<0.25 (84% of points in the coarse−mode validation), attributable to a marine−dominated aerosols scene, the EE is 51%, whereas for AOD≥0.25 (16% of points in the coarse−mode validation), attributable to dust−dominated aerosols, the EE is significantly lower than 1 sigma."
have been rephrased as:
"However, for the coarse−mode, the MAIAC validation shows an MBE of –0.08 and 46% of points within the envelope of EE=±(0.1$AOD$ + 0.05)) from (Falah et al., 2021), significantly lower with respect to the other two classes. In case of AOD<0.25 (84% of points in the coarse−mode validation), attributable to a marine−dominated aerosols scene (Toledano et al., 2007), 51% of points are within the EE, whereas for AOD≥0.25 (16% of points in the coarse−mode validation), attributable to dust−dominated aerosols (Rogozovsky et al., 2023; Bodenheimer et al., 2021; Toledano et al., 2007), the % of points within EE is significantly lower than 1 sigma (i.e. 68% of points falling in the EE envelope)."

Line 200: Could you provide any relevant reference for the limit use here for marine and dust dominated aerosol scenes?
The limit used has been extrapolated from Bodenheimer et al. (2021) who recommend a threshold of AOD > 0.3 for dust dominated scene (chosen also in Rogozovsky et al, 2023), while they report AOD levels below 0.2 and 0.3<AE<0.6 for a combination of marine and anthropogenic aerosols from Toledano et al. (2007). The references have been added to the paper as indicated in in a previous comment (see Lines 199 & 201 comment).

Bodenheimer, S, Nirel, R, Lensky, IM, Dayan, U. The synoptic skill of aerosol optical depth and angstrom exponent levels over the Mediterranean Basin. Int J Climatol. 2021; 41: 1801–1820. https://doi.org/10.1002/joc.6931

Rogozovsky, I., Ohneiser, K., Lyapustin, A., Ansmann, A., and Chudnovsky, A.: The impact of different aerosol layering conditions on the high-resolution MODIS/MAIAC AOD retrieval bias: The uncertainty analysis, Atmos. Environ., 309, 119930, https://doi.org/10.1016/j.atmosenv.2023.119930, 2023.

Toledano, C., Cachorro, V.E., Berjon, A., de Frutos, A.M., Sorribas, M., de la Morena, B.A. and Goloub, P. (2007), Aerosol optical depth and Ångström exponent climatology at El

Arenosillo AERONET site (Huelva, Spain). Q.J.R. Meteorol. Soc., 133: 795-807. https://doi.org/10.1002/qj.54

Lines 203-205: Could you elaborate or rephrase this sentence because it is a little bit confusing?
The sentence "(Qin et al., 2021) show as MAIAC regional background models seem to be affected by local aerosol properties as they are retrieved by AERONET climatology, suggesting that, further improvements are needed in case of coarse−mode dominated classification."
has been modified as follows taking into account also the Referee#2 Ln. 205 suggestion:
"(Rogozovsky et al., 2023; Qin et al., 2021) show that the MAIAC algorithm is sensible to the aerosol size. (Rogozovsky et al., 2023) observed that the underestimation of MAIAC compared to AERONET is related to the presence of dust (characterized by high depolarization ratio and low AE). This result suggests that, further improvements are needed in case of coarse−mode dominated classification."

Line 278: Instead of "the extra" better "the different AOD levels" since lower values of AOD have been found for the local scale?
Done.

Line 293: Could you please quantify here most of the time?
The following sentence has been added to the paper: "… for most of the cities considered in our study (58.8%) larger than the regional AOD."

Lines 301-302: I would recommend changing the order of factors explaining the negative LTRR values, starting from inhomogeneity of AOD at the regional domain.
Lines 310-313: Another recommendation is to add here as reason of inhomogeneity in AOD at the regional domain the spatial extent of the city (also related to topography, but not always), different location of sources like industrial areas etc. which in combination with the meteorological conditions could lead to different spatial patterns inside the 100 x 100km^2 domain.
The above comments (Lines 301-302 and Lines 310-313) have been taken into account to rephrase the Lines 300-313 as follows:
"On the contrary, negative LTRR values indicate a lower local city AOD than the regional one, suggesting a possible inhomogeneity in AOD within the rather large (100 x 100 km$^2$) regional domain since the observed negative LTRR values were in general very small (in the order of some %). This could be true especially for coastal sites, or partly mountainous sites, where topography plays an important role. Furthermore, this inhomogeneity may be related to i) the spatial extent of the city, which may impact the AOD levels of the regional scale, ii) the different location of emission sources, such as the location of industrial areas, which, combined with favorable meteorological conditions can lead to inhomogeneous spatial patterns in the regional domain.

An alternative explanation to negative LTRR values would be local aerosol loss at urban scale. However, systematic urban loss processes are not easy to identify. Sedimentation and dry deposition processes are not expected to be particularly enhanced over urban areas, nor is precipitation, compared to its regional surrounding. On the other hand, the urban heat island with increased temperatures could lead to evaporation of particles. For instance, (Pirhadi et al., 2020) finds that due to its semi-volatile character, about 50% of ambient PM2.5 aerosol mass evaporated when heated up in a thermo−denuder from ambient temperature (~13°C in winter, 23°C in summer and up to 50°C). The urban heat island effect depends on the size and additional heat production within an urban area. It is restricted to light wind meteorological

conditions and it is more pronounced during night, while MAIAC observations are made during daytime. For these reasons, we consider that evaporation of semi–volatile aerosol under higher urban temperatures could only play a limited effect in our dataset. In the frame of the present analysis it is in general difficult to distinguish between these two loss and inhomogeneity effects."

Lines 338-339: Repetition with what is inside parenthesis in lines 337-338
The sentence "Barcelona shows the largest LTRR ($0.57 \pm 0.02$) among the cities studied" has been removed accordingly.

Line 360: The 0.83 value here is the coefficient of determination $R^2$ of the regression and this value is different from Pearson correlation coefficient r=0.89 shown in figure 5?
The r=0.89 is the Pearson correlation coefficient. The value of r=0.83 was given by error and this has been corrected in the revised paper.

Line 1021: I recommend here to add "… cities (not metropolitan regions) with more than 1 million…". It is something that is also different in the Eurostat database.
Thank you for this suggestion, it has been implemented.

**Technical corrections**

Line 986: In Fig.2 explain red lines and dashed green lines in a, c,d, e.
The following sentences have been added to the caption of Fig. 2: "Red solid lines represent the straight lines passing through the origin with 2, 1 and 0.5 slope coefficients respectively. The green dotted line represents the regression line."

Line 990: In Fig.2 "times x"? Expressed in percentage you mean?
Yes, the value is expressed as a percentage. This aspect has been clarified in the main manuscript at line 1021.

Line 990: In Fig.2 "EE" instead of "EE5"
Done. The Fig. 2 and its caption have been changed accordingly.

Line 1009: In Fig 5. please explain solid and dashed lines and it is messing AOD nest to local versus regional.
The following sentence has been added to the caption of Fig. 5: "Solid and dashed lines represent the 1:1 and regression lines respectively."

Line 1006: Perhaps "mean AOD" better than "mean of the boxplot"?
"the mean of the boxplot" has been replaced by "the AOD mean".

**Reviewer #2**

Specific suggestions:

Remove "one" from the last sentence of the Abstract.
Done.

Ln. 35: Replace "still huge" with "large remaining"
Done.

Ln. 38: Replace "are everyday exposed to significant aerosol levels" with "experience a significant particulate matter exposure"
Done.

Ln. 42: Correct to " and last for several consecutive days"
Done.

Ln. 43: Delete "If such episodes occur frequently". Visibility and air quality deteriorate even if it happens once.
Done.

Ln. 48: "to potentially"
Done.

Ln. 51: "and is a matter"
Done.

Ln. 53 Re-phrase "properties". AOD alone does not give much information about aerosol properties.
Yes, the reviewer is right. The word "properties" has been removed and the text rephrased as follows: "The Aerosol Optical Depth (AOD) is a key parameter to investigate aerosol load and distribution."

Ln. 68: Please, add reference to Hammer et al., 2020 (Global Estimates and Long-Term Trends of Fine Particulate Matter Concentrations (1998–2018)
The indicated reference has been added in the manuscript.

Ln. 72: remove "aerosol"
Done.

Ln. 84: "and distinguish between smoke and dust scenes." Add reference to Lyapustin et al., 2012 (Lyapustin, A., S. Korkin, Y. Wang, B. Quayle, and I. Laszlo, 2012b: Discrimination of biomass burning smoke and clouds in MAIAC algorithm, *Atmos. Chem. Phys.*, 12, 9679–9686, doi:10.5194/acp-12-9679-2012.)
Done.

Ln. 86: I recommend adding the latest reference to most comprehensive DB, DT, MAIAC and NOAA AOD comparison and validation analysis (Su, X., M. Cao, L. Wang, X. Gui, M. Zhang, Y. Huang, Y. Zhao, Validation, inter-comparison, and usage recommendation of six latest

VIIRS and MODIS aerosol products over the ocean and land on the global and regional scales, *Science of The Total Environment*, v. 884, 2023, 163794, https://doi.org/10.1016/j.scitotenv.2023.163794)

Following the referees suggestion, we added this reference.

Ln. 90: I recommend adding two important missed references: a) van Donkelaar, A., et al. (2021). Monthly Global Estimates of Fine Particulate Matter and Their Uncertainty. *Environmental science & technology*, doi: 10.1021/acs.est.1c05309, and b) Wei, J. et al., 2021: Reconstructing 1-km-resolution high-quality PM2.5 data records from 2000 to 2018 in China: spatiotemporal variations and policy implications. *Remote Sensing of Environment*, 2021, 252, 112136. https://doi.org/10.1016/j.rse.2020.112136

Following the referees suggestion, we added these two references.

Ln. 95: "In the Paris area"
Done.

Ln. 96: Change to "aims to achieve"
Done.

Ln. 122: Replace "taking into account available observations in the day" with "using available cloud-free observations"
Done.

Ln. 130: Replace "was chosen large enough in order to avoid effects due to the city and its plume" with "was chosen large enough to minimize effects of city's pollution"
Done.

Ln. 205: A similar analysis and conclusion over Israel was just published in (Irina Rogozovsky, Kevin Ohneiser, Alexei Lyapustin, Albert Ansmann, Alexandra Chudnovsky, The impact of different aerosol layering conditions on the high-resolution MODIS/MAIAC AOD retrieval bias: The uncertainty analysis, Atmospheric Environment, V. 309, 2023, 119930, https://doi.org/10.1016/j.atmosenv.2023.119930.
We added this reference. Please see also the reply to Referee#1 Lines 203-205.

Ln. 435: "indicates an average local–to– regional ratio of 39%," Please add "local-to-regional excess ratio" to indicate that the ratio >1.
Thank you for this suggestion.